# Aberrant Sialylation in Cancer: Biomarker and Potential Target for Therapeutic Intervention?

**DOI:** 10.3390/cancers13092014

**Published:** 2021-04-22

**Authors:** Silvia Pietrobono, Barbara Stecca

**Affiliations:** Tumor Cell Biology Unit, Core Research Laboratory, Institute for Cancer Research and Prevention (ISPRO), Viale Pieraccini 6, 50139 Florence, Italy

**Keywords:** cancer, sialyltransferases, invasion, metastasis, angiogenesis, cell death, survival, immune evasion, biomarker, inhibitors

## Abstract

**Simple Summary:**

Sialylation is a post-translational modification that consists in the addition of sialic acid to growing glycan chains on glycoproteins and glycolipids. Aberrant sialylation is an established hallmark of several types of cancer, including breast, ovarian, pancreatic, prostate, colorectal and lung cancers, melanoma and hepatocellular carcinoma. Hypersialylation can be the effect of increased activity of sialyltransferases and results in an excess of negatively charged sialic acid on the surface of cancer cells. Sialic acid accumulation contributes to tumor progression by several paths, including stimulation of tumor invasion and migration, and enhancing immune evasion and tumor cell survival. In this review we explore the mechanisms by which sialyltransferases promote cancer progression. In addition, we provide insights into the possible use of sialyltransferases as biomarkers for cancer and summarize findings on the development of sialyltransferase inhibitors as potential anti-cancer treatments.

**Abstract:**

Sialylation is an integral part of cellular function, governing many biological processes including cellular recognition, adhesion, molecular trafficking, signal transduction and endocytosis. Sialylation is controlled by the levels and the activities of sialyltransferases on glycoproteins and lipids. Altered gene expression of these enzymes in cancer yields to cancer-specific alterations of glycoprotein sialylation. Mounting evidence indicate that hypersialylation is closely associated with cancer progression and metastatic spread, and can be of prognostic significance in human cancer. Aberrant sialylation is not only a result of cancer, but also a driver of malignant phenotype, directly impacting key processes such as tumor cell dissociation and invasion, cell-cell and cell-matrix interactions, angiogenesis, resistance to apoptosis, and evasion of immune destruction. In this review we provide insights on the impact of sialylation in tumor progression, and outline the possible application of sialyltransferases as cancer biomarkers. We also summarize the most promising findings on the development of sialyltransferase inhibitors as potential anti-cancer treatments.

## 1. Introduction

Glycosylation is a post-translational modification that involves transfer of glycosyl moieties to specific amino acid residues of proteins to form glycosidic bonds through the activity of glycosyltransferases. The most widely occurring cancer-associated changes in glycosylation include sialylation, fucosylation, O-glycan truncation, and N- and O-linked glycan branching [1]. Sialylation consists in the addition of sialic acid to the terminal position of glycan chains on glycoproteins and glycolipids. Aberrant sialylation is one of the most common change in glycosylation occurring in cancer and it is dynamically regulated by two sets of enzymes, sialyltransferases (STs, responsible for sialylation) and sialidases (NEUs, involved in desialylation). Hypersialylation can be the effect of upregulation of sialyltransferases, downregulation of sialidases or a combination of both [2], and results in an excess of negatively charged sialic acid on the cell surface. Sialic acid accumulation contributes to immune evasion as well as reduced efficacy of chemotherapy and radiotherapy [3,4]. In addition, it can promote tumor metastasis by stimulating migration, invasion and angiogenesis [5].

Several key reviews have thoroughly described the role of aberrant glycosylation and sialylation in cancer [5,6,7]. In this review, we summarize recent advances on the regulation of ST expression in cancer and explore the mechanisms by which STs promote cancer progression through activation of invasion and metastasis, stimulation of angiogenesis, drug resistance, immune evasion and tumor cell survival. In addition, we provide insights into the possible use of sialyltransferases as cancer biomarkers and review the most promising findings on the development of sialyltransferase inhibitors.

## 2. Sialyltransferases

The family of human sialyltransferases consists of 20 enzymes that transfer sialic acid from cytidine monosphosphate N-acetylneuraminic acid (CMP-Neu5Ac) to the terminal glycosyl group of various glycoproteins and glycolipids. Sialic acid plays a crucial role in several cellular interactions, including with the extracellular matrix, epithelial cells, immune cells, antibodies and other intercellular processes. The attachment of sialic acid to the underlying glycan chain can occur through different glycosidic linkages (α-2,3, α-2,6, α-2,8). Sialyltransferases can be classified into four main groups depending on the types of glycosidic bonds: ST3GAL1-6 (α-2,3-sialyltransferases), ST6GAL1-2 and ST6GALNAC1-6 (α-2,6-sialyltransferases) and ST8SIA1-6 (α-2,8-sialyltransferases) [8].

Sialyltransferases are type II transmembrane glycoproteins usually located in the Golgi apparatus. STs are expressed in a tissue-specific manner and each presents substrate specificity, although with some degree of redundancies [9]. STs share also a conserved protein structure, that consists of a short N-terminal cytoplasmic domain, a transmembrane domain (TMD), a stem region of variable length and a catalytic domain. The latter contains four conserved sialylmotifs, namely ‘L’ (long), ‘S’ (short), ‘III’ (third position in sequence) and ‘VS’ (very small), which are involved in recognition of donor and acceptor substrates and catalytic activity [10,11].

## 3. Regulation of Sialyltransferase Gene Expression

Sialyltransferase expression appears to be regulated mainly at the level of transcription by a number of factors, including oncogenes, transcription factors (TF), miRNAs, long noncoding RNAs (lncRNA), hormones and natural compounds.

In the ST3GAL family, gene transcription results in distinct mRNAs, which are generated by alternative splicing and alternative promoter usage, resulting in tissue-specific expression. The human ST3GAL1 gene contains nine exons. Results of site-directed mutagenesis indicated that the Sp1 binding sites and an upstream stimulatory factor 1 (USF) binding site in the promoter are involved in the transcriptional regulation of human ST3GAL1 [12]. Human ST3GAL2 has two isoforms regulated by promoters P1 and P2 [13]. Human ST3GAL3 has only one mRNA and an Sp1 element [14], whereas several transcripts and promoters have been described for human ST3GAL4, ST3GAL5 and ST3GAL6 [15].

The proto-oncogene c-Myc has been reported to regulate transcription of ST3GAL1, 2 and 5 in colon cancer cells [16]. In hormone-sensitive prostate cancer (PC) cells, androgens regulate ST3GAL2 transcription by inducing promoter demethylation and increasing GD1a expression, a sialogangloside associated with tumor progression [17]. In breast cancer cells, prostaglandin E2 (PGE2), one of the final products of the cyclooxygenase-2 (COX-2) pathway, can induce ST3GAL1 expression in both ER-positive and ER-negative cell lines [18]. In hepatocellular carcinoma (HCC) cell lines, the tumor suppressors miR-26a, miR-548l and miR-34a have been shown to negatively regulate the expression of ST3GAL5 [19]. In addition, miR-26a negatively regulated ST3GAL6, inducing the suppression of HCC cell proliferation, migration, and invasion in vitro [20].

A recent study showed that in melanoma cells the oncogenic TFs SOX2 and GLI1 co-regulate ST3GAL1 transcription. Chromatin immunoprecipitation assay showed co-occupancy of both TFs at a distal enhancer element located at about 10 kb upstream ST3GAL1 transcription start site. Consistently, authors demonstrated that ST3GAL1 mediates SOX2- and GLI1-induced melanoma invasiveness [21]. This study suggests that regulatory regions other than promoters are involved in transcriptional regulation of ST3GAL1.

On the other hand, in renal cell carcinoma (RCC) cells ST3GAL1 is positively regulated by MEG3, a lncRNA with a tumor suppressor activity. MEG3 interacts with the transcription factor c-Jun, which negatively regulates ST3GAL1 transcription by directly binding to ST3GAL1 promoter. Accordingly, ST3GAL1 and MEG3 expression positively correlate, with higher expression of both factors in adjacent tissues compared to RCC [22]. Hu et al. demonstrated that the lncRNA ST3GAL6 antisense 1 (ST3GAL6-AS1), which derives from the promoter region of ST3GAL6, recruits histone methyltransferase MLL1 to the promoter region of ST3GAL6, inducing activation of ST3GAL6 transcription, and consequent inhibition of the PI3K/AKT signaling in colorectal cancer (CRC) cell lines. According with the anti-tumorigenic function of the ST3GAL6-AS1/ST3GAL6 axis, the expression of both ST3GAL6-AS1 and ST3GAL6 is down-regulated in CRC tissues compared to adjacent tissues [23].

In the ST6GAL family, ST6GAL1 is ubiquitously expressed and is the most investigated ST in cancer, while ST6GAL2 is mainly expressed in the brain. Modulation of ST6GAL1 expression is the result of the regulation of a single gene by at least three promoters (P1, P2 and P3), which generate at least three major mRNA isoforms in a tissue-specific manner through differential assemblage of 5′ untranslated regions [24]. ST6GAL1 is a good example of sialyltransferase directly controlled by oncogenes. N-Ras and H-Ras have been shown to promote ST6GAL1 transcription through the RalGEF effector [25]. Furthermore, ST6GAL1 is negatively regulated by the tumor suppressor RUNX3 [26]. According to Dorsett and colleagues the TF SOX2 promotes the expression of ST6GAL1 in ovarian cancer cells by binding a region proximal to the P3 promoter [27]. The tumor suppressor miR-9 has been shown to inhibit the expression of ST6GAL1, resulting in reduced invasiveness of mouse HCC cells by inhibition of α-2,6-linked sialylation and integrin-β1/FAK signaling pathway [28]. ST6GAL1 is also negatively regulated by miR-214 in colorectal cancer cells [29].

Transcription of members of the ST6GALNAC family is regulated by specific proteins and miRNAs. For instance, the homeobox TF CDX2 positively regulates the expression of ST6GALNAC1 in gastric carcinoma cell lines and human intestinal metaplasia by binding CDX-binding sites close to the start codon [30]. In prostate cancer cells, androgen stimulation can induce the expression of a novel alternative splice variant of ST6GALNAC1, which encodes a shorter protein isoform that is still fully functional as a sialyltransferase and able to induce expression of the sialyl-Tn (sTn) antigen [31]. Further studies showed that miR-182/miR135b negatively regulate ST6GALNAC2 expression through the PI3K/AKT pathway to mediate the invasiveness as well as resistance of colorectal cancer cells to 5-fluorouracil (5-FU) [32]. In addition, miR-4299 modulates the invasive ability of human thyroid carcinoma cells by targeting ST6GALNAC4 in vitro and in vivo [33].

The ST8SIA family can be regulated by miRNAs. Ma and colleagues demonstrated that miR-26a and miR-26b negatively regulate ST8SIA4 via directly targeting the 3′-UTR of ST8SIA4 to inhibit migration and invasion in vitro of aggressive breast cancer cells [34]. Another report showed that ST8SIA4 can be inhibited by miR-146a and miR-146b in follicular thyroid carcinoma cells [35]. Recently, the lncRNA ST8SIA6-AS1, which functions as a potent tumor promoter in several types of cancer, has been shown to be transcriptionally repressed by the tumor suppressor p53 [36]. ST8SIA can also be upregulated by natural compounds. For instance, curcumin induces transcriptional activation of ST8SIA1 in A549 human lung carcinoma cells [37], physcion modulates transcription of ST8SIA6 in human neuroblastoma cells [38], whereas ST8SIA3 is upregulated by retinoic acid treatment in U87MG glioblastoma cells [39].

## 4. Roles of Sialyltransferases in Cancer

### 4.1. Sustaining Proliferation and Tumor Growth

An essential feature of cancer cells is the ability to proliferate even in absence of proliferative stimuli. Sialylation has been shown to affect proliferative signaling pathways, although the mechanistic link between increased sialylation and cellular proliferation may be exclusive to specific types of cancer.

One of the first evidence that the ST3GAL family is involved in cancer came from the finding that ST3GAL1 overexpression promotes mammary tumorigenesis, although the mechanism by which this ST exerts its oncogenic function was not elucidated [40]. Few years later a study reported that the GDNF (glial cell derived neurotrophic factor) receptor alpha (GFRA1), a coreceptor that recognizes the GDNF family of ligands, is a substrate of ST3GAL1. Silencing of ST3GAL1 decreased GDNF-induced phosphorylation of RET, AKT and ERα, and reduced GDNF-mediated breast cancer cell proliferation. In turn, GDNF induced transcription of ST3GAL1 through AKT/Sp1, revealing a positive feedback loop between ST3GAL1 and GDNF/GFRA1 signaling in breast cancer cells [41].

The ST6GAL family has been implicated in regulating cancer cell proliferation. ST6GAL1 silencing attenuated proliferation and colony formation ability of PC-3 and DU145 prostate cancer cell lines, through inhibition of the PI3K/AKT/GSK-3β/β-catenin pathway [42]. Similarly, ST6GAL2 silencing inhibited MCF-7 and T47D breast cancer cell proliferation by arresting cell cycle progression at G0/G1 phase and reducing the fraction of cells in S phase [43].

A further study showed that ST6GALNAC1 silencing reduces proliferation and clonogenicity of mouse hepatocarcinoma cell lines by inhibiting the PI3K/AKT/NF-κB pathway [44]. Likewise, knockdown of ST8SIA4 repressed MDA-MB-231 cell proliferation in vitro and tumor growth in vivo, as demonstrated by the reduction of Ki67-positive cells in the tumor tissue [34].

Recent reports have highlighted the role of the lncRNA ST8SIA6-AS1 in promoting cancer cell proliferation. Indeed, ST8SIA6-AS1 is upregulated in HCC and can facilitate cell proliferation and resistance to cell apoptosis via sponging miR-4656 and upregulation of HDAC11 expression, suggesting an important role of the ST8SIA6-AS1/miR-4656/HDAC11 axis in regulating HCC cell proliferation [45]. Consistently, downregulation of ST8SIA6-AS1 suppressed cell proliferation in vitro and restrained HCC tumorigenesis in vivo [46]. A similar role is played by lncRNA ST8SIA6-AS1 in breast cancer through the p38 MAPK signaling pathway [47]. Recently, depletion of ST8SIA6-AS1 has been shown to cause mitotic catastrophe, massive apoptosis, and cell cycle arrest in a broad spectrum of cancer cell lines [48].

In contrast to the examples reported above, in gliomas increased expression of STs may have a tumor inhibitory effect. In glioblastoma cell lines, DNA methylation within core promoter regions leads to transcriptional silencing of ST6GAL1, whose expression can be reactivated by 5-aza-2′-deoxycytidine [49]. Consistently, intracranial injection of U373 MG glioma cells overexpressing ST6GAL1 or ST3GAL3 led, respectively, to a complete lack or drastic reduction in tumor growth in vivo [50]. Similarly, glioma cells showed very low expression of ST6GALNAC5, which when overexpressed into U373 MG cells inhibited glioma growth in vivo [51] (Figure 1; Table 1). It is striking that increase in sialyltransferase expression promotes pro-tumorigenic and pro-metastatic effects in the majority of cancer types, whereas it can have opposite effects on cancers originating from neural tissues, such as glioma. This could be due to the fact that polysialylation plays a key role in adult brain plasticity and regeneration [52].

### 4.2. Activating Invasion and Metastasis, and EMT Inducing Events

The ability of cancer cells to invade and spread is essential in the development of an invasive phenotype. This is a multistep process starting with local invasion of cancer cells into the surrounding tissues, followed by intravasation, survival in the environment of the circulatory system, extravasation and growth in the new tissue [53]. The active role of STs in supporting the acquisition of invasive and metastatic phenotypes is well established in numerous types of cancer [7]. Changes in sialylation have been implicated in mediating epithelial-mesenchymal transition (EMT), a reversible cellular phenotype switching characterized by loss of epithelial markers in favor of a migratory mesenchymal state. EMT is the main driving factor of malignant tumor metastasis [54].

The link between ST3GAL and increased invasion and metastasis is well established. ST3GAL1 has been shown to promote migration and invasion of the metastatic HCC cell line HCCLM3 in vitro and its expression is associated with poor prognosis in human HCC [55]. In addition, overexpression of ST3GAL1 can promote migration and peritoneal dissemination of ovarian cancer cells via the EGFR signaling [56]. A further report showed that ST3GAL1 plays a crucial role in TGF-β1-induced EMT in ovarian cancer. Treatment of ovarian cancer cells with TGF-β enhanced the expression of ST3GAL1, leading to a decrease in E-cadherin levels and increase of N-cadherin and vimentin [57].

A recent study reported that ST3GAL1 expression correlates with melanoma progression and highlighted the critical role of ST3GAL1 in driving melanoma metastasis. Silencing of ST3GAL1 suppressed melanoma migration and invasion, and reduced the ability of aggressive melanoma cells to enter the bloodstream, colonize distal organs and survive in the metastatic environment, without affecting melanoma cell proliferation nor orthotopic tumor growth in vivo. Mass spectrometry and functional assays revealed that ST3GAL1 explicates the pro-invasive functions through the receptor tyrosine kinase AXL, and that ST3GAL1-mediated sialylation is able to induce AXL dimerization and activation [21].

Other members of the ST3GAL family have shown to regulate migration and invasion. For instance, ST3GAL3 modulated breast cancer cell adhesion and invasion by inducing the expression of invasion-related molecules, including β1 integrin, matrix metalloproteinase (MMP)-2, MMP-9 and COX-2 [58]. ST3GAL3 and ST3GAL4 have been shown to promote pancreatic cancer cell adhesion, motility and migration in vitro and to enhance metastatic potential in vivo [59,60]. Consistently, downregulation of either ST3GAL3 or ST3GAL4 decreased pancreatic cancer cell migration, invasion and E-selectin-dependent adhesion [61]. Gomes et al. found that expression of ST3GAL4 in MKN45 gastric cancer cells resulted in enhanced synthesis of the sLex antigen and an increased invasive phenotype both in vitro and in vivo through the activation of c-Met [62]. Likewise, increased α-2,3 sialylation by ST3GAL4 led to increased metastatic potential of gastric cancer cells and correlates with poor prognosis [63].

Glavey and colleagues demonstrated that high expression of ST3GAL6 is associated with reduced survival in multiple myeloma (MM) patients. ST3GAL6 silencing resulted in a significant decrease in α-2,3–linked sialic acid levels on the surface of MM cells, with a reduction of adhesion to MM bone marrow stromal cells and diminished transendothelial migration in vitro. ST3GAL6-silenced MM cells displayed reduced homing and engraftment in vivo, with decreased tumor burden and prolonged survival [64]. Furthermore, Hu et al. showed that the lncRNA ST3GAL6-AS1, which is transcribed from the promoter region of ST3GAL6 in opposite orientation, decreases CRC invasion in vitro and in vivo and inhibits the PI3K/AKT signaling, resulting in Foxo1 nuclear translocation. In turn, ST3GAL6-AS1 is regulated by Foxo1, establishing a ST3GAL6-AS1-PI3K/AKT-Foxo1 positive loop which contributes to restrain CRC progression [23].

Recent reports highlighted the importance of ST6GAL1 activity in driving EMT. For instance, in breast cancer cells overexpression of ST6GAL1 promoted TGFβ-induced EMT through down-regulation of E-cadherin-mediated cell adhesion and up-regulation of integrin-mediated cell migration [65]. In pancreatic cancer cells, high expression of ST6GAL1 led to increased α-2,6 sialylation and activation of EGFR, as well as upregulation of mesenchymal markers and enhanced cell invasiveness. Interestingly, the EGFR inhibitor erlotinib attenuated the ST6GAL1-dependent differences in EGFR sialylation and activation, EMT marker expression and invasiveness [66]. In addition, increase of α-2,3 sialylation induced by ST6GAL1 promoted integrin α5β1-dependent adhesion in hepatocarcinoma cells [67].

ST6GAL1 has been shown to promote a more migratory and invasive phenotype in several types of cancer. In prostate cancer cells, ST6GAL1 silencing inhibited cell migration and invasion through downregulation of the PI3K/AKT/GSK-3β/β-catenin pathway [42]. Similarly, in non-small cell lung cancer (NSCLC) cells, ST6GAL1 silencing suppressed migration and invasion in vitro and in vivo, through inactivation of the Notch1/Hes1/MMPs pathway [68]. In ovarian cancer cells, α-2,6 sialylation of fibroblast growth factor receptor 1 (FGFR1) by ST6GAL1 activated the extracellular signaling-regulated protein kinase 1/2 (ERK1/2) and focal adhesion kinase (FAK) pathway, enhancing cell migration and drug chemoresistance [69].

Although most of the current studies recognize the tumor promoting role of ST6GAL1, few reports suggest opposite function in colorectal cancer. Indeed, ST6GAL1 expression is decreased at advanced stage of CRC, with a significantly higher expression at stages I and II (non-metastatic) and lower expression at stages III and IV (metastatic) compared with normal tissues [70]. Jung et al. found that migration and invasion were enhanced in ST6GAL1-depleted SW620 colon cancer cells in vitro, and that ST6GAL1 silencing promoted liver metastasis in vivo by reducing the cellular pool of the metastasis suppressor KAI1 [71]. Similarly, ST6GAL1 overexpression inhibited the metastatic ability of CRC in vitro and in vivo by stabilizing intercellular adhesion molecule 1 (ICAM-1) [72].

In contrast, a recent report pointed to a tumor promoting role of ST6GAL2 in breast cancer. In a study of 633 breast cancer patients, ST6GAL1 expression was associated with tumor stage, survival and estrogen receptor (ER)/progesterone receptor (PR)/human epidermal growth factor receptor 2 (HER2) status. Consistently, silencing of ST6GAL2 negatively affected cancer progression by inhibiting cell adhesion and invasion, and reducing the expression of ICAM-1, VCAM-1, CD24, MMP2, MMP9, and CXCR4 [43].

Ozaki et al. showed that overexpression of ST6GALNAC1 in gastric cancer cells induced expression of the sTn antigen, one of the most well-known cancer-associated glycan structures. ST6GALNAC1 overexpression led to enhanced intraperitoneal metastasis and shortened survival, effects that were alleviated by administration of anti-sTn mAb. This study identified the glycoproteins MUC1 and CD44 as the major carrier proteins for sTn, suggesting an involvement of these glycoproteins in the acquisition of a metastatic phenotype by gastric cancer cells [73]. Consistently, another report showed that ST6GALNAC1 silencing inhibits expression of sTn, resulting in reduced gastric cancer cell migration and invasion in vitro. Intraperitoneal administration of ST6GALNAC1 siRNA-liposome inhibited peritoneal dissemination of MKN45 cells and prolonged survival in mice. Mechanistically, suppression of ST6GALNAC1 caused a significant reduction in the expression of IGF-1 mRNA accompanied by reduced STAT5b phosphorylation [74]. Fujita et al. used a ST6GALNAC1-expressing MDA-MB-231 cell line and, contrarily to their expectations, found that sTn expression impaired adhesion of breast cancer cells to bone marrow stromal cells, fibronectin and type I collagen, attenuating the development of osteolytic bone lesions [75]. Another study showed that silencing of ST6GALNAC1 suppresses migration and invasion of hepatocarcinoma cells through PI3K/AKT/NF-κB pathway [44].

Using an in vivo functional RNA interference metastasis screen, Murugaesu and colleagues identified ST6GALNAC2 as a breast cancer metastasis suppressor gene [76]. ST6GALNAC2 silencing altered the profile of O-linked glycans on cell surface, increasing soluble lectin galectin-3 binding and tumor cell clustering at metastatic sites, leading to increased metastatic burden. This study provides strong evidence that high ST6GALNAC2 levels in the ER-negative breast cancers are associated with improved overall survival and that low expression of ST6GALNAC2 might be used as a biomarker in ER-negative breast cancer to identify patients who may benefit from treatment with galectin-3 inhibitors [76]. Conversely, two studies showed that ST6GALNAC2 silencing inhibits the invasive ability of MDA-MB-231 breast cancer cells and of FTC-238 follicular thyroid cancer cells in vitro through the PI3K/AKT/NF-κB signaling pathway [77,78]. The opposite effects of ST6GALNAC2 silencing may be ascribed to different roles of ST6GALNAC2 in vitro and in vivo, or during the initial and late stages of the metastatic process.

Studies correlating the ST8SIA family with tumor progression are mainly related to ST8SIA1 and ST8SIA4. Sarkar and colleagues demonstrated that ST8SIA1 (GD3 synthase) modulates EMT through activation of c-Met and is required for breast cancer cell migration, invasion and metastasis formation in vivo [79]. A further study shed light on the role of ST8SIA1 in triple negative breast cancers (TNBC), showing that ST8SIA1 is highly expressed in primary TNBC, and its knockdown inhibits orthotopic xenograft growth of TNBC cells and abolishes lung metastases [80]. Mass spectrometry analysis for N-glycan profiling in breast cancer tissues and cell lines identified ST8SIA4 as one of the most differentially expressed genes in MDA-MB-231 breast cancer cells compared to non-tumor MCF-10A cells. Expression of ST8SIA4 was associated with cancer metastasis, whereas its knockdown repressed MDA-MB-231 cell migration and invasion [34]. Conversely, the same group found that ST8SIA4 was downregulated in highly invasive follicular thyroid carcinoma (FTC) cells and tissues, and that ST8SIA4 inhibited migration and invasion of (FTC) cells both in vitro and in vivo, suggesting that ST8SIA4 downregulation may contribute to FTC progression [35].

Recently, the lncRNA ST8SIA6-AS1 has been shown to promote migration and invasion in breast cancer cells through the p38 MAPK signaling [47], in HCC cells by regulating the expression of the Cul4-associated factor 4-like 2 (DCAF4L2) [46] and in lung adenocarcinoma cells by sponging miR-125a-3p to increase nicotinamide N-methyltransferase (NNMT) expression [36] (Table 2).

### 4.3. Promoting Immune Evasion

Cancer cells must be able to avoid detection and destruction by the immune system to efficiently metastasize and spread throughout the body. It has been known for a long time that aberrant glycosylation of cancer cells protect them from destruction by the immune system [4]. In particular sialic acid on the surface of cancer cells is believed to play a crucial role in immune modulation and tumor immune evasion, and sialic acid blockade could provide a therapeutic approach to creates an immune-permissive tumor microenvironment [81]. Accordingly, reduction of sialic acid on the murine B16 melanoma cells by knocking down the sialic acid transporter Slc35A1 has shown to reduce the presence of α2,6-linked sialic acids on the cell surface, to enhance effector T cell response and to increase influx and activity of natural killer (NK) cells, reinforcing anti-tumor immunity and curtailing tumor growth [82].

In addition, hypersialylated glycans on both glycoproteins and glycolipids on the surface of cancer cells are recognized by Siglecs, sialic acid-binding immunoglobulin-type lectins found on the surface of immune cells. The interaction of sialic acid on cancer cells with Siglecs can promote immunosuppressive cues, conferring protection to tumor cells [3]. For instance, tumor cell death mediated by natural killer (NK) cells is hampered by interactions between NK-expressing Siglec-7 or Siglec-9 and sialylated glycans on tumor cells [83,84]. Accordingly, targeting Siglec-7 and Siglec-9 is under investigation as a therapeutic approach to enhance NK cell response against cancer cells [84]. Another report showed that glioma cells evade myeloid-derived suppressor cells by expressing ligands for Siglec-3, Siglec-5, Siglec-7 and Siglec-9 [85]. Furthermore, binding of a cancer-specific MUCIN-1 glycoform to Siglec-9 expressed by primary macrophages induced a tumor-associated macrophage (TAM) phenotype that promotes disease progression [86]. Barkal and colleagues found that the highly sialylated cell-surface protein CD24, which is overexpressed in ovarian and breast cancers, binds to Siglec-10 on macrophages, protecting tumor cells from phagocytosis. Genetic ablation of CD24 or Siglec-10, as well as blockade of the CD24-Siglec-10 interaction, restored phagocytosis. Consistently, CD24 blockade reduced tumor growth and increased survival in a MCF-7 cell xenograft mouse model [87]. Wang and colleagues identified Siglec-15 as a critical immune suppressor, which is broadly upregulated on human cancer cells and tumor infiltrating myeloid cells. Siglec-15 suppressed antigen-specific T cell responses in vitro and in vivo. Genetic ablation or antibody blockade of Siglec-15 restores anti-tumor immunity and inhibits tumor growth in vivo [88]. A recent report showed that cancer-derived sialylated IgG in the tumor microenvironment can promote tumor immune escape by binding to Siglecs on effector CD4+ and CD8+ T cells [89].

Only recently the involvement of specific STs in immune evasion of cancer cells started to be elucidated. Singh and Choi showed that in lymph nodes melanoma cells bind to subcapsular sinus macrophages that express Siglec-1, resulting in colonization of the nodes. Knockout of ST3GAL3 in melanoma cells reduced α-2,3 sialylation and the metastatic ability of these cells [90]. Furthermore, ST6GAL1 has been shown to promote immune escape in hepatocarcinoma cells by enhancing levels of CD147, MMP9, MMP2 and MMP7, and inhibiting T cell proliferation [91]. Lin and colleagues recently demonstrated that ST3GAL1-mediated O-linked sialylation of CD55, an important complement regulatory protein, acts like an immune checkpoint molecule for breast cancer cells to evade immune attack. Consistently, inhibition of ST3GAL1 has been proposed as a strategy to block CD55-mediated immune evasion. Mechanistically, authors showed that O-linked desialylation of CD55 by ST3GAL1 silencing results in increased C3 deposition and complement-mediated lysis of breast cancer cells, enhancing sensitivity to antibody-dependent cell-mediated cytotoxicity [92]. A recent work showed that circulating monocytes that infiltrate the tumor can interact with tumor-derived sialylated structures through the receptors Siglec-7 and Siglec-9 in pancreatic ductal adenocarcinoma. Authors identified ST3GAL1 and ST3GAL4 as the main contributors to the synthesis of ligands for Siglec-7 and Siglec-9 in tumor cells. Tumor-derived sialic acids force monocytes to produce IL-10 and IL-6 and subsequently drive their differentiation to immune suppressive TAMs through the activation of the Siglec-9 receptor [93] (Table 3).

### 4.4. Evading Apoptosis and Cell Death

To develop and strive in the tumor microenvironment cancer cells need to evade and overcome cellular apoptosis. Sialylation has been associated with the programmed cell death of several cell types. The TNF family of death receptors (TNFRs) regulate programmed cell death, which includes TNFR1, DR4, DR5, and Fas (CD95), represents a category of molecules that is commonly disrupted in human tumors and has been strongly implicated in tumor cell survival [94].

In a colon cancer model, Swindall and colleagues identified the Fas receptor (FasR) as an ST6GAL1 substrate. ST6GAL1-mediated sialylation of FasR blocked Fas internalization and the formation of the death-inducing signaling complex, disabling apoptotic signaling in colon cancer cells [95]. ST6GAL1 exerted a similar effect on tumor necrosis factor (TNF)-induced cell death by sialylation of the TNF receptor 1 (TNFR1), which inhibits its internalization, preventing the induction of apoptosis and promoting cell survival in pancreatic and ovarian cancer cells [96]. This study provides a novel mechanism by which ST6GAL1 may promote tumor cell survival within TNF-rich inflammatory tumor microenvironments. Furthermore, upregulation of ST6GAL1 was shown to enhance resistance to TNF-mediated apoptosis in gastric cancer-derived organoids, suggesting a role for ST6GAL1 in epithelial cell longevity of gastric adenocarcinoma [97].

Another study investigated the role of O-glycosylation of cell surface death receptors in terms of the sensitivity to TRAIL-induced apoptosis, and found that expression of normal extended O-glycans can enhance sensitive to TRAIL through O-glycosylated DR4 or DR5. This represents the first molecular mechanistic insight into how expression of truncated O-glycans, such as Tn/sTn antigens, alters signaling in tumor cells to promote their survival [98] (Table 4).

### 4.5. Inducing Angiogenesis

The formation of neovasculature is necessary for the supply of nutrients and oxygen during tumor development and metastasis formation. Hypoxia is the key factor that promotes tumor angiogenesis. Hypoxic tumor cells secrete vascular endothelial growth factor A (VEGFA), which binds to VEGF receptor 2 (VEGFR2) that is expressed on neighboring vascular endothelial cells, promoting tumor angiogenesis [99].

The expression of STs is closely related to angiogenesis. A recent study suggested that ST3GAL1 may affect angiogenesis through vasorin [100]. Authors reported that ST3GAL1 silencing reduces tumor growth with a remarkable decrease in vascularity of MCF7 xenografts and identified the angiogenin vasorin (VASN) as the main substrate of ST3GAL1 in breast cancer cells. ST3GAL1 silencing or desialylation of VASN enhanced its binding to TGF-β1, reducing TGF-β1 signaling and angiogenesis, as indicated by impaired tube formation, suppressed angiogenesis gene expression and reduced activation of Smad2 and Smad3 in HUVEC cells [100]. The down-regulation of ST6GAL1 has been shown to interfere with the transduction of PECAM-VEGFR2 and integrin-β3 in mouse Lewis lung carcinoma cells, leading to endothelial cell apoptosis, which impairs tumor angiogenesis [101]. In line with this, knockdown of ST6GAL1 in a osteosarcoma cell line reduced levels of VEGF [102], suggesting a link between aberrant sialylation and angiogenesis also in this cancer type.

Hypoxia plays a key role in promoting the formation of new vessels through the activation of several pro-angiogenic factors [103]. A recent report showed that ST6GAL1 increases HIF-1α accumulation in ovarian and pancreatic cancer cells grown in a hypoxic environment, and induced the expression of HIF-1α transcriptional targets, including the glucose transporters GLUT1 and GLUT3 and the glycolytic enzyme PDHK1. Interestingly, cells grown in hypoxia for several weeks displayed increased ST6GAL1 expression [104]. These results point toward the establishment of a positive regulatory loop between ST6GAL1 and HIF-1α, and suggest that under hypoxic conditions ST6GAL1 confers pro-survival and pro-angiogenetic features. Furthermore, hypoxia has been shown to affect the sialylation profile of ovarian cancer and TNBC cell lines. Indeed, hypoxia-exposed ovarian cancer and TNBC cells displayed enhanced EMT features and migration with increased levels of ST3GAL4 mRNA [105]. A seminal paper showed that cancer cells can undergo hypoxia-induced glycan remodeling which can confer tumors resistance to anti-VEGF treatment. Thus, targeting glycosylation-dependent lectin-receptor interactions may increase the efficacy of anti-VEGF treatment [106]. Conversely, overexpression of ST8SIA1 has been shown to inhibit survival of pancreatic cancer cells, to downregulate the expression of angiogenesis regulatory proteins and to inhibit EGF/VEGF-driven angiogenic cell growth [107] (Table 5).

### 4.6. Promoting Chemoresistance

One key malignant phenotype of cancer cells is their ability to cope with the presence of therapeutics. Hypersialylation of cancer cells has been shown to promote chemoresistance in several types of cancer. This feature has been studied particularly in relation to ST6GAL1, which can induce direct sialylation and activation of EGFR, FGFR1 and HER2. In colon cancer cells, silencing of ST6GAL1 increased the anti-cancer effect of the EGFR inhibitor gefitinib, whereas ST6GAL1 overexpression decreased the cytotoxic effect of gefitinib [108]. Similar results were observed in ovarian cancer cells by Britain et al., providing the first evidence that sialylation of EGFR by ST6GAL1 promotes EGFR activation and consequent resistance to gefitinib-mediated cell death [109]. The same group showed that ST6GAL1 confers resistance to cisplatin in ovarian tumor cells, although the mechanism underlying this effect was not elucidated [110]. A recent study showed that ST6GAL1 overexpression induces α-2,6-sialylation of FGFR1 and that high levels of ST6GAL1 decrease the anticancer effect of the FGFR1 inhibitor PD173047 in ovarian cancer cells [69]. Liu et al. identified HER2 as an ST6GAL1 substrate and showed that HER2 α-2,6 sialylation confers protection against trastuzumab-mediated apoptosis through AKT and ERK1/2 pathway in gastric cancer cells [111]. Chakraborty et al. showed that ST6GAL1 promotes chemoresistance in pancreatic ductal adenocarcinoma by abrogating gemcitabine-mediated DNA damage [112]. A recent report showed that ST6GAL1 protects also against radiation-induced gastrointestinal damage in mice [113].

The ST3GAL family is also involved in promoting chemoresistance. ST3GAL1 silencing increased sensitivity to a combination of RET inhibitor vandetanib and ER inhibitor tamoxifen in ERα-positive breast cancer cells [41]. In ovarian cancer cells, overexpression of ST3GAL1 increased resistance to paclitaxel, whereas downregulation of ST3GAL1 produced the opposite effect [57]. In chronic myeloid leukemia (CML) cells, ST3GAL1 was shown to mediate multidrug resistance (MDR). Indeed, ST3GAL1 is significantly upregulated in CML without MDR compared to CML with MDR. Consistent with these results, ST3GAL1 silencing induced MDR in KCL22 cells, whereas chemoresistance of KCL22/ADR cells is decreased upon ST3GAL1 overexpression [114]. Similarly, ST3GAL3 silencing sensitizes ovarian cancer cells to cisplatin-induced apoptosis [115]. Furthermore, overexpression of ST3GAL4 and ST3GAL6 make gastric adenocarcinoma MKN45 spheroids more resistant to crizotinib [116], a drug that targets the receptor tyrosine kinases MET and RON.

Another report elucidated the role of sialylation in regulating cancer (MDR) in acute myeloid leukemia (AML). Sialylation is involved in the development of MDR of AML cells likely through ST3GAL5 and ST8SIA4, with high expression of ST3GAL5 in drug-sensitive cells and high expression of ST8SIA4 in adriamycin-resistant cells. Suppression of ST3GAL5 or ST8SIA4 overexpression regulate the activity of PI3K/Akt signaling and increase the expression P-glycoprotein (P-gp) and MDR-related protein 1 (MRP1) in HL60 cells, contributing to the development of MDR in AML cells [117]. Furthermore, ST8SIA1 has been shown to be upregulated in chemoresistant TNBC patients and its inhibition sensitizes TNBCs to chemotherapy through suppression of Wnt/β-catenin and FAK/Akt/mTOR pathway [118].

In the ST6GALNAC family, overexpression of ST6GALNAC1 was shown to induce sTn expression and to confer resistance to cisplatin or 5-fluorouracil (5-FU) in MKN45 gastric cancer cells, whereas its knockdown restored galectin-3-binding sites and sensitized tumor cell to drug-induced cell death [119]. Furthermore, ST6GALNAC2 silencing reversed the chemoresistance of HCT-8 and LoVo colorectal cancer cells to 5-fluorouracil (5-FU) induced by inhibition of miR-135b or miR-182 [120] (Table 6).

### 4.7. Enhancing Stemness

Several studies demonstrated that sialylation is essential for the establishment and maintenance of stem cell pluripotency. A significant change in protein sialylation levels has been reported during differentiation, with higher levels of ST6GAL1 in the undifferentiated human pluripotent stem cells (PSCs) compared to the non-pluripotent cells. Consistently, ST6GAL1 silencing decreased the efficiency of somatic cell reprograming [121]. In addition, it has been reported that α-2,3 sialylation regulates the stability of the neural stem cell marker CD133 [122].

A recent study provides another key mechanism through which ST3GAL1, a major invasiveness factor, impacts tumorigenic growth of glioblastoma by regulation the stem cell/progenitor pool [123]. Authors demonstrated that ST3GAL1 is triggered by the TGFβ signaling pathway in the mesenchymal patient cohort and regulates brain tumor formation through APC/C-Cdh1–targeted control of FoxM1 protein degradation. This work provides evidence that ST3GAL1 sustains self-renewal in vitro and tumor growth of glioma-propagating cells (GPCs), promoting the expression of the neurodevelopmental factors OLIG2, SALL2 and SOX2. Conceivably, ST3GAL1 depletion increased the activity of the APC/C-Cdh1 complex as cells enter mitosis, which then targets the N-terminal of FoxM1 protein for subsequent degradation, thus arresting GPCs at the G2/M phase with consequent cell differentiation [123].

Schultz and colleagues reported that the sialyltransferase ST6GAL1 plays a crucial role in driving a cancer stem cell (CSC) state in ovarian and pancreatic cancers [124]. Using complementary assays, authors demonstrated that ST6GAL1 confers CSC characteristics, including spheroid growth, chemotherapy resistance, and tumor-initiating potential. Furthermore, ST6GAL1 induced expression of two key tumor-promoting transcription factors, SOX9 and Slug, also known to play crucial roles in the maintenance of stem/progenitor pools [124]. A recent study showed that ST6GALNAC1 contributes to the maintenance of colorectal cancer CSCs by activating the AKT pathway in cooperation with galectin-3 [125].

Nguyen et al. showed that ST8SIA1 regulates breast cancer stem-like cells (BCSC) in TNBC. Indeed, knockout of ST8SIA1 completely inhibited in vitro BCSC functions, including mammosphere formation. Interestingly, ST8SIA1 positively regulated the expression of several breast cancer stem-like cells (BCSC)-associated genes, such as BCL11A, FOXC1, CXCR4, PDGFRβ and SOX2. Proteomic analysis revealed that FAK and its downstream AKT-mTOR signaling are activated by ST8SIA1 in BCSCs [80] (Table 7).

## 5. Sialylated Glycoproteins and Sialyltransferases as Cancer Biomarkers

Several sialylated proteins have been approved as cancer biomarkers, including prostate-specific antigen (PSA) used for screening of PC, α-fetoprotein used for diagnosis and monitoring of HCC, CA125 that detects the mucin glycoprotein MUC16 in ovarian cancer and thyroglobulin for thyroid cancer [2]. A recently proposed new biomarker is a glycan called sialylated tumor-related antigen (sTRA), which allows the detection of chemotherapy-resistant pancreatic cancers [126]. The use of sTRA holds important implications for patients with resistant pancreatic ductal adenocarcinoma (PDAC) to predict relapse after surgery and guide the choice of treatments.

ST6GAL1 might have a prognostic role in ovarian carcinoma, where its increased expression correlates with reduced patient survival [124]. In another ovarian cancer study (*n* = 517), Wichert et al. found that high ST6GAL1 mRNA levels significantly correlated with lymphovascular invasion and shorter survival, whereas high ST6GAL1 protein expression was associated with advanced stage, distant metastasis and shorter recurrence-free intervals. This study provided evidence that ST6GAL1 expression might help to identify cases with high risk of chemoresistance and metastatic relapse [127]. In addition, ST6GAL1 has been recently proposed as a novel tumor-derived secreted biomarker that identifies lenvatinib-susceptible FGF19-driven highly malignant HCCs [128].

A study with 286 patients showed that ST3GAL1 is an independent adverse prognostic factor for recurrence and survival of patients with clear cell renal cell carcinoma (ccRCC) [129]. A transcriptomic analysis of 185 glycogenes identified ST3GAL6 as a biomarker that predicts clinical outcome in urinary bladder cancer (UBC). ST3GAL6 was negatively associated with the subtype with luminal feature in UBC patients (*n* = 2130 in total) and increased ST3GAL6 expression was positively correlated to tumor stage, grade, and survival in UBCs (*n* = 52) [130].

A further expression analysis of soluble E-selectin and five sialyltransferases (ST3GAL1-4, ST6GAL1) in 135 surgically treated node-negative breast cancer patients, revealed that a high ST3GAL3/ST6GAL1 ratio and high levels of E-selectin correlate with a poor prognosis for both relapse-free and overall survival [131]. A recent report analyzed the relationship between the α-2,6 sialyltransferases ST6GAL1, ST6GAL2, and ST6GALNAC1 and tumor-infiltrating lymphocyte (TIL) in different breast cancer molecular subgroups. It was found that an increase in tumor infiltrating lymphocytes (TIL) was associated with low expression of ST6GAL1 in epidermal growth factor receptor 2-overexpressing (HER2) breast cancers and by high expression of ST6GAL2 in triple-negative breast cancers [132]. Furthermore, elevated ST3GAL2 and ST3GAL3 mRNA expression is associated with advanced stages of oral squamous cell carcinoma with lymph node involvement, suggesting a role of these two sialyltransferases in oral squamous cell carcinoma progression [133].

In a recent glycoproteomic study, the ST6GALNAC2 gene was found highly upregulated across all oncogenic cell lines investigated, compared with other glycosylated proteins, including sialyltransferases [134]. However, levels of ST6GALNAC2 mRNA and protein are downregulated in colorectal cancers compared to adjacent noncancerous tissues [32]. Therefore, further studies are needed to establish the potential for ST6GALNAC2 as a cancer biomarker. A further report identified ST6GALNAC3 as a target of aberrant promoter hypermethylation in 705 prostate cancer tissues and detected circulating tumor DNA (ctDNA) for ST6GALNAC3 in liquid biopsies of 27 PC patients [135].

Human cells differ from many other mammal cells by the lack of the sialic acid N-glycolylneuraminic acid (Neu5Gc) and abundance of its precursor N-acetylneuraminic acid (Neu5Ac). Humans are not able to synthesize Neu5Gc due to an exon deletion in the enzyme cytidine monophosphate-N-acetylneuraminic acid hydroxylase gene [136]. However, Neu5Gc is metabolically incorporated into human tissues from dietary sources (particularly red meat), and detected at even higher levels in some human cancers [137]. Most humans produce anti-Neu5Gc antibodies, which have been proposed as cancer biomarkers. High pre-existing anti-Neu5Gc IgG levels in the plasma were associated with increased risk of colorectal cancer in a cohort of 71 colorectal cancer cases and matched controls [138]. This association is promising and warrants confirmation in a larger cohort.

## 6. Sialyltransferase Inhibitors

In the last two decades, several sialyltransferase inhibitors have been identified by design, in natural products or microbial metabolites, and from high-throughput screening methods. Most of these inhibitors can be classified into: (i) acceptor analogues, (ii) donor analogues, based on the structure of CMP-Neu5Ac, (iii) bisubstrate analogues, and (iv) transition-state analogues, whereas others can be obtained from natural products. Whilst many of these inhibitors are not good candidates for therapeutic intervention due to their high polarity and charge that counteract cell absorption, some of them showed encouraging effects in vivo.

### 6.1. Acceptor Analogues

Sialyltransferases catalyze the transfer of a sialic acid residue from a sugar nucleotide donor to a glycoconjugated acceptor. Both the nature of the nucleotide donor and the terminal structure of the glycan portion of the sugar acceptor determine the specificity of each sialyltransferase. However, reports on acceptor-type inhibitors are limited.

The Hashimoto group synthesized several analogues of methyl N-acetyl-β-lactosaminide from lactose, and identified the 6′-deoxy analogue as the first acceptor-analogue inhibitor with remarkable inhibitory activity toward α-2,6-sialyltransferases [139]. Okazaki et al. also described a series of carba-oligosaccharides possessing a similar structure to the three-dimensional conformation of natural activated oligosaccharides, but unable to be metabolized by glycosidases [140]. Among them, the imino-linked methyl 5a′-carba-β-lactoside has been shown to possess a potent and specific inhibitory activity toward rat recombinant α-2,3-sialyltransferase [140].

Recently, Gupta and colleagues investigated the different acceptor specificity for individual α-2,3-sialyltransferases, demonstrating that ST3GAL1, 2 and 4 prefer Galβ1, 3GalNAc (Type-III) substrates, with ST3GAL1 also sialylating Galβ1,3(GlcNAcβ1,6)GalNAc reactive groups [141]. Conversely ST3GAL3, 4 and 6 act both on Galβ1,3GlcNAc (Type-I) and Galβ1,4GlcNAc (Type-II) glycans, suggesting that a broad analysis of the structure-activity relation could represent a promising strategy for the design of structure-specific inhibitors.

### 6.2. Donor Analogues

Over the last decade, several groups focused on the design and synthesis of CMP-sialic acid analogues (donor analogs), able to prevent sialic acid transfer by competing with the natural donor CMP-Neu5Ac in the binding to the active site of sialyltransferases. These small molecules mostly contain a cytidine fragment that was found to be critical for the binding to the enzyme.

#### 6.2.1. Nucleoside Fragment (Cytidine Analogues)

Previous work by Kajihara et al. reported CMP-9″-modified-Neu5Ac analogues bearing the phenyl-substituted alkyl-amide group as efficient inhibitors of rat liver α-2,6-sialyltransferase [142]. Miyazaki and colleagues reported that methylcytosine derivatives of cytidine monophosphate (CMP) showed selective inhibition of ST3GAL3 and ST3GAL4 activity, but not of ST6GAL1, possibly due to the different environment in the active site from an enzyme subtype to another [143]. CMP-Neu5Ac analogues were also shown to exert competitive inhibition against ST8SIA2, blocking ST8SIA2-mediated polysialylation of cancer cells and reducing in vitro migration [144].

A study from Li et al. described a more potent CMP-Neu5Ac analogue, in which a cyclopentane α-hydroxyphosphonate was coupled with cytidine phosphoramidite, mimicking the planar structure of the donor in the transition state [145]. The authors demonstrated that this small molecule exerts a strong inhibitory activity against ST6GAL1, proposing the cyclopentanoid-type compounds as novel sialyltransferase inhibitors as biological probes or drug leads [145].

More recently, Montgomery and colleagues developed a series of uridine-based compounds in which the α-hydroxyphosphonate was coupled with a 5′-amino-5′-deoxyuridine fragment instead of a cytidine [146]. The authors reported no cytotoxicity and higher potency of these compounds in inhibiting ST6GAL1 activity compared to their ancestors [146].

#### 6.2.2. Sugar Fragment (Sialic Acid Analogues)

The first synthesized 2-deoxy-2-fluorosugar nucleotide was proposed to act as a competitive inhibitor versus nucleotide sugar donor substrates, with selective inhibition properties for both fucosyl- and sialyl-transferases [147]. However, its reduced ability to go across cell membrane strongly limits its application in vivo.

Paulson and co-workers reported the development of a cell-permeable, peracetylated analogue of natural occurring Neu5Ac, named 2,4,7,8,9-pentaacetyl-3Fax-Neu5Ac-CO2Me (P-3Fax-Neu5Ac), characterized by a fluorine atom proximal to the endocyclic oxygen of the sialic acid backbone that can inhibit sialyltransferase function and hence sialylation [148]. P-3Fax-Neu5Ac is converted into CMP-3F-NeuAc, an extremely poor substrate for sialyltransferases because of the electron-withdrawing effect of the fluorine atom, with a consequent loss in synthesis of sialoglycoproteins and a feedback inhibition of the sialic acid biosynthesis pathway [148]. P-3Fax-Neu5Ac has been reported as a global sialyltransferase inhibitor in vivo, producing liver dysfunction due to the absence of terminal sialylation on galactose residues of serum glycoproteins that engulfs their normal clearance [149]. This pan-sialyltransferase inhibitor has been shown to readily cause depletion of α-2,3- and α-2,6-linked sialic acids, and to strongly reduce in vitro cell adhesion, migration and in vivo growth of murine melanoma cells [150]. Its formulation into poly(lactic-co-glycolic acid) nanoparticles induced long-term sialic acid blockade, and strongly prevented metastasis formation in a murine lung metastasis model of melanoma [151]. Natoni et al. also reported the efficacy of P-3Fax-Neu5Ac in multiple myeloma both in vitro and in vivo. This compound has been shown to affect post-translational modifications on α4β1 and α4β7 integrins, preventing their interactions with the receptors E-selectin, MADCAM1 and VCAM1 [152]. This impairs extravasation and retention of multiple myeloma cells in the bone marrow, where they are generally protected from chemotherapeutics such as bortezomib. As a matter of fact, administration of P-3Fax-Neu5Ac has been showed to improve survival by enhancing bortezomib sensitivity in a mouse model of multiple myeloma [152], suggesting that this targeting strategy to interfere with aberrant sialylation is expected to be useful in reducing the severity of the disease.

Modifications to the sialic acid backbone of P-3Fax-Neu5Ac to generate a C-5-carbamate 3-fluoro sialic acid analogue has been shown to improve the inhibitory potency of the lead compound due to a more efficient metabolism to its CMP analogue, and to determine a long-lasting inhibition of both α-2,3- and α-2,6-sialyltransferase activity in murine melanoma cells [153].

### 6.3. Bisubstrate Analogues

Inhibitors of the bisubstrate analogue type are designed to mimic donor and acceptor substrates, containing motifs of both donor and acceptor that are covalently bound to each other. A first study from Hinou et al. reported bisubstrate-type analogues having CMP-NeuAc and lactose moieties connected by an alkanedithiol linker as potent inhibitors of both α-2,3 and α-2,6 sialyltransferase activity [154]. Later, Izumi and collaborators described a bisubstrate analogue containing a cytidin-5′-yl sialylethylphosphonate instead of CMP−Neu5Ac as donor substrate, and a galactose as the acceptor one, tethered with a phosphodiester moiety [155]. This compound demonstrated high potency against the activity of rat ST3GAL1 and ST6GAL1 in vitro [155]. However, the potential of bisubstrate analogues as sialyltransferase inhibitors needs further investigations.

A recent study by Seibel and co-workers aimed to describe the contribution of different residues composing the binding sites of both donor CMP-Neu5Ac and acceptor Galβ1-3GalNAc-R to the catalysis by ST3GAL1 [156]. Structural modelling of human ST3GAL1 by its porcine homolog led to the identification of a series of residues required for optimal donor and/or substrate binding (N147, N170, Y191 and Y230), whose occurrence appears critical for hST3GAL1 activity [156]. Authors suggested that clarification of protein-ligand molecular interactions with both donors and acceptors is expected to aid the design of potent and selective inhibitors toward sialyltransferases.

### 6.4. Transition-State Analogues

The mechanism of sialyltransferases involves the nucleophilic attack of a deprotonated hydroxyl of an acceptor on the anomeric carbon of Neu5Ac, which generates an oxocarbenium-like transition-state (TS), with the CMP moiety acting as a leaving group. Previous studies proposed TS analogues as potent sialyltransferase inhibitors [157,158]. These compounds mimic the planar oxocarbenium-like TS of the sialyltransferase mechanism and use the CMP-Neu5Ac donor as a starting point. Several of these compounds were generated by addition of a planar bond to the sialic acid mimic and replacement of the carboxylate group with phosphonate, and showed improved inhibitory activity [159]. However, most of these inhibitors have been designed with limited data on target structure of sialyltransferases, and their charged phosphodiester linkage is associated with poor cell permeability and low bioavailability.

The recent characterization of the crystal structure of several mammalian sialyltransferases provided insight on the mode of action of sialyltransferases and allowed the use of computational tools to support structure-based design of inhibitors with favorable pharmacokinetic properties.

The first crystal structure was reported by Rao et al. [160]. The crystalized enzyme is a porcine ST3GAL1 variant, possessing about 85% homology with the corresponding human enzyme. By structural and kinetic characterization of pST3GAL1, authors established the basis for understanding specificity and for the design of selective inhibitors toward individual sialyltransferases [160].

More recently, using the human ST6GAL1 structure [161], Montgomery and co-workers performed molecular docking and molecular dynamics simulations to investigate the effect of phosphodiester-linker replacement with a neutral isostere, such as a rigid 1,2,3-triazole or a flexible carbamate [162]. Authors reported that the interaction of these modified-linkers with hST6GAL1 is slightly more favorable than that of the phosphodiester-linker [163], providing for the first time insight into the binding of these compounds to the catalytic site of ST6GAL1 and suggesting targeting of this site as a conceivable route to design selective ST6GAL1 inhibitors. Likewise, Guo et al. reported that the introduction of an amide bond to mimic the oxocarbenium ion in the TS coupled with substitution of Neu5Ac with aromatic or aliphatic rings led to a series of potent inhibitors with high selectivity for ST6GAL1 [164].

More recently, a computational analysis allowed to determine the affinity and selectivity of different potential TS inhibitors for the binding site of the crystallized human ST8SIA3 [165]. The authors reported that substitution of the phosphodiester-linker with a more flexible carbamate better fitted the binding site of ST8SIA3, whilst the replacement of cytidine nucleotide with uridine appeared more selective against ST6GAL1, as the cytidine moiety is supposed to be crucial for ST6GAL1 activity [165], suggesting a distinctive selectivity of the derivatives for the binding to different sialyltransferases. Thus, a thorough characterization of binding affinities based on free energy calculations could be useful to develop potent and more selective sialyltransferase inhibitors. All these studies provide a rationale for the development of selective TS-based inhibitors, as each sialyltransferase subfamily is expected to differ within this region to accommodate the different sialyl-acceptors.

### 6.5. Natural Products

Although natural products have been used for hundreds of years, their application still represent a great resource for drug discovery, particularly as antibacterial or anti-tumor agents.

#### 6.5.1. Soyasaponin I

Soyasaponin I (SsaI) is an amphiphilic oleanane triterpenoid from soybean, endowed with antimicrobial, anti-inflammatory, anticarcinogenic and cardiovascular-protective activities. SsaI was found to selectively reduce α-2,3-linked sialic acid expression without affecting other surface glycans, acting as a competitive inhibitor affecting CMP-Neu5Ac binding to ST3GAL1 (Ki = 2.1 μM) [166]. SsaI was also shown to enhance the adhesion ability of breast cancer cells in vitro [167] and to significantly decrease the migratory and metastatic abilities of murine melanoma cells both in vitro and in vivo [168].

#### 6.5.2. Steroidal Compounds

Among the steroidal compounds, epiandrosterone succinyl ester, a potent glutathione S-transferase inhibitor isolated from Schistosoma japonicum, has been reported to inhibit ST3GAL1 activity [169]. Despite efforts to develop synthetic analogues of epiandrosterone with alternations of the succinyl ester unit to enhance its potency, these compounds showed reduced or absent inhibitory effects compared to the parental one [169].

A potent steroidal derivative is represented by lithocholic acid, a substrate of nuclear pregnane X receptor which mimics the pentacyclic ring of SsaI. This compound has been shown to exert non-competitive inhibition of CMP-Neu5Ac likely by binding the ST3GAL1 active site, resulting in decreased enzyme activity [169]. The design of sixteen different synthetic derivatives of lithocholic acid highlighted the importance of the carboxylic acid group of lithocholic acid for promoting the binding affinity to ST3GAL1. Indeed, reduction of the carboxylic acid to the corresponding alcohol, or the replacement of a hydroxyl group with a ketone moiety significantly decreased potency of the compound. Conversely, replacement of the primary alcohol with a 1,4-disubstituted 1,2,3-triazole ligand strongly improved binding affinity [169]. Among these derivatives, Lith-O-Asp was shown to exhibit low micromolar inhibitory activity toward α-2,3 (ST3GAL1 and ST3GAL3) and α-2,6 (ST6GAL1) sialyltransferases [170]. Treatment with Lith-O-Asp was reported to strongly reduce in vitro migration ability and to prevent metastasis formation in vivo of murine breast cancer cells via inhibition of integrin sialylation and FAK/paxillin signaling, and upregulation of anti-angiogenic factors [170].

Another Lith analogue developed by the same group, AL-10, was found to be cell-permeable and able to selectively inhibit α-2,3-sialyltransferase activity. Treatment of lung cancer cells with AL-10 strongly impaired cell adhesion via disruption of cytoskeleton organization, and almost completely abrogated migration and invasion by decreasing sialylation of integrins α5, αV and β1 [171]. Furthermore, AL-10 exhibited a potent anti-metastatic activity in vivo by suppressing lung metastasis without affecting liver and kidney function of experimental animals, making it a promising candidate for treatment of metastatic cancer [171].

Recently, Fu and colleagues employed the “LCA extension strategy”, in which the cyclopentane ring side chain of Lith-O-Asp was extended with one or more ethylene glycol units to generate isozyme-specific sialyltransferase inhibitors [172]. Within the second-generation ST inhibitors, FCW34 exhibited good pharmacokinetic characteristics and a great permeability potency [172]. FCW34 was shown to reduce breast cancer cell migration by suppressing α2,3-N-ST3GAL3-catalyzed N-glycoprotein sialylation of β-integrins. Also, FCW34 was found to inhibit tumor growth and metastasis in a mouse model of triple negative breast cancer, and to suppress angiogenic activity in transgenic zebrafish models [172], suggesting that it could represent a reasonable therapeutic approach for sialyltransferases-driven breast cancer.

#### 6.5.3. Flavonoids

Flavonoids belong to a class of natural polyphenols widely present in plants, endowed with anti-inflammatory, anti-microbial, anti-oxidant and anti-tumor effects. By chemical synthesis of flavonoid derivatives, Hidari et al. obtained successful inhibition of α-2,3 and α-2,6 sialyltransferase activity with IC50 less than 10 μM [173]. A double bond formation between C2 and C3 positions on the C-ring and the presence of hydrophilic groups on the B-ring appeared crucial for the inhibitory activity of flavonoids. The authors reported a unique mechanism of inhibition with mixed inhibition kinetics of these compounds, although it is unclear whether flavonoids interact with the sialylmotif L or S acting as CMP-NeuAc competitors, or with other amino-acid residues inducing conformational changes in the substrate binding site [173]. Synthetic derivatives of epigallocatechin gallate, the main flavonoid compound of green tea, such as 5,7-dideoxy-epigallocatechin gallate (DO-EGCG), were previously reported to moderately inhibit rat brain ST3GAL1, but it also showed inhibitory activity toward fucosyltransferases with the same potency [174].

#### 6.5.4. Alginate Oligosaccharide

Alginate oligosaccharide (AOS) is a degradation product of alginate, a polysaccharide found in several brown seaweeds, with antioxidant effects. It consists of a mannuronic acid and a guluronic acid linked via 1,4-glycosidic bonds. AOS has been recently shown to inhibit the expression of ST6GAL1 in human prostate cancer cells via the Hippo/YAP pathway, leading to altered expression profile of α-2,6-linked sialic acids [175]. Importantly, this compound showed no apparent cytotoxicity in normal human cells, highlighting its potential as a promising anti-cancer strategy.

#### 6.5.5. Ginsenosides

Ginsenosides are a class of natural steroid glycosides and triterpene saponins found almost exclusively in the Panax ginseng plant. Huang and collaborators recently demonstrated that ginsenosides block sialylation of α-2,3- and α-2,6-linked sialic acids in human liver cancer cell lines (HEPG2). Through a molecular docking simulation, the authors found that these compounds negatively interact with both ST3GAL1 and ST6GAL1 [176].

### 6.6. Others

Meril et al. recently reported a chimeric antigen receptor (CAR) T-cell approach to target cancer-associated glycosylation patterns [177]. As hypersialylated proteins could be recognized by sialic acid-binding immunoglobulin-type lectins (Siglecs), the authors tuned genetic engineering of T-cells expressing Siglec-based CARs with different chimeric receptors based on the exodomain of human Siglec-7 and Siglec-9 molecules, to enable the recognition and elimination of tumor cells. These S7 CAR or S9 CAR showed a significant antitumor activity in vitro against several cancer cell lines derived from tumors of breast, cervix, lung, pancreatic, prostate and melanoma, and, most importantly, a strong tumor growth delay in a xenograft mouse model of human melanoma [177].

In another study, Yamanaka et al. identified putative inhibitors of the Ganglioside GM3 Synthase (GM3S), known as ST3GAL5, by high-throughput screening (HTS) of a chemical library. These compounds revealed a mixed mode of inhibition, likely through the binding to the allosteric pockets of GM3S [178]. Although the selectivity and the lipophilicity of these compounds need to be improved, they might have a great potential to be developed into selective sialyltransferase inhibitors.

## 7. Conclusions

It is clear from the presented literature that ST3GAL1 and ST6GAL1 are the most extensively studied sialyltransferases in cancer. Therefore, there is still a lot to do to deepen our understanding on the entire panel of 20 human sialyltransferases and their different roles in cancer progression.

One of the major roles of STs is promoting tumor progression by supporting cell migration and invasion, cancer cell survival, as well as avoiding detection and destruction by the immune system. Regarding the latter feature, further investigation is needed to elucidate the involvement of specific STs in immune evasion. Other important features that need to be further explored are the mechanisms by which STs promote drug resistance. In addition, much efforts should be put into the development of novel glycoproteomic approaches to study sialylation in cancer cells.

The rationale for targeting STs in cancer is supported by the many studies discussed in this and other reviews, particularly to enable the design of effective anti-metastatic agents that target key sialyltransferase subtypes, with the potential to prevent metastatic spread. However, many challenges remain to be addressed in this field of research. First, to proceed to clinical trial it is critical that sialyltransferase inhibitors are subtype selective to reduce off-target effects on the liver and kidney. In addition, the development of clinically relevant sialyltransferase inhibitors as potential anti-cancer treatments requires good cell permeability and bioavailability. Among the several classes of ST inhibitors, the CAR T-cell approach to target cancer-associated glycosylation patterns and the possibility to block specific Siglec in immune cells deserve particular attention. The recent reports of crystal structures for pig ST3GAL1, rat and human ST6GAL1, and human ST8SIA3 have enabled the use of structure-based design to drive the design of ST inhibitors. Based on that, in the near future efforts need to be directed towards the discovery and development of subtype-selective and clinical relevant ST inhibitors.

## Figures and Tables

**Figure 1 cancers-13-02014-f001:**
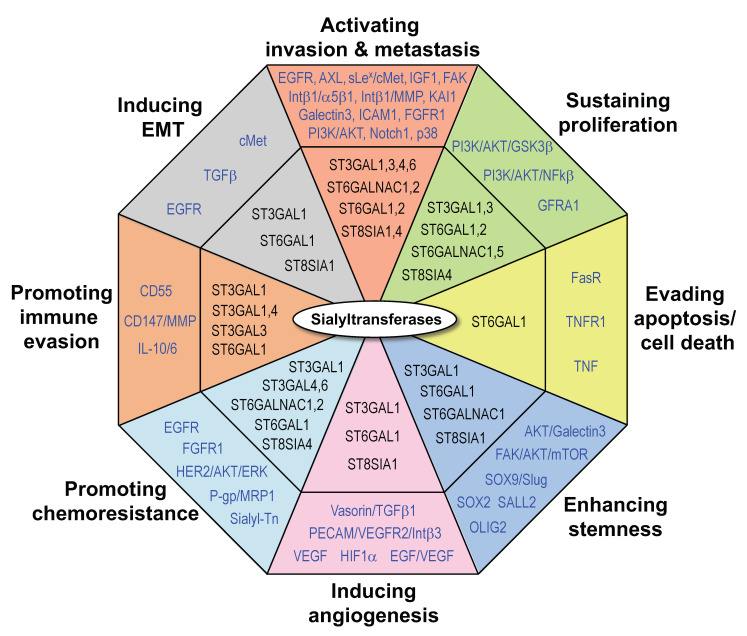
Roles of sialyltransferases in cancer. Indicated are STs (in black) and their proposed targets or downstream mediators (in blue). Abbreviations used in the figure are listed in the “Abbreviation” section.

**Table 1 cancers-13-02014-t001:** Sialyltransferases implicated in cancer cell proliferation.

STs	Target/Mediator	Effect	Cancer Type	References
ST3GAL1	GFRA1	Promotion	Breast cancer	[41]
ST3GAL3/ST6GAL1		Inhibition	Glioma	[50]
ST6GAL1	PI3K/AKT/GSK3β/βcatenin	Promotion	Prostate cancer	[42]
ST6GAL2		Promotion	Breast cancer	[43]
ST6GALNAC1	PI3K/AKT/NF-kB	Promotion	HCC	[44]
ST6GALNAC5		Inhibition	Glioma	[51]
ST8SIA4		Promotion	Breast cancer	[34]
ST8SIA6-AS1	miR-4656/HDAC11	Promotion	HCC	[45]
ST8SIA6-AS1	p38	Promotion	Breast cancer	[47]

**Table 2 cancers-13-02014-t002:** Sialyltransferases involved in invasion and metastasis.

STs	Target/Mediator	Effect	Cancer Type	References
ST3GAL1	TGF-β1	Promotion	Ovarian cancer	[57]
ST3GAL1	EGFR	Promotion	Ovarian cancer	[56]
ST3GAL1	AXL	Promotion	Melanoma	[21]
ST3GAL1		Promotion	HCC	[55]
ST3GAL3	Integrinβ1/MMP-2/9	Promotion	Breast cancer	[58]
ST3GAL3/4		Promotion	Pancreatic cancer	[59,60,61]
ST3GAL4	sLe^x^/cMet	Promotion	Gastric cancer	[62]
ST3GAL4		Promotion	Gastric cancer	[63]
ST3GAL6		Promotion	Multiple myeloma	[64]
ST3GAL6-AS1	PI3K/AKT-Foxo1	Inhibition	Colorectal cancer	[23]
ST6GAL1	Integrin α5β1	Promotion	HCC	[67]
ST6GAL1	Integrin β1/FAK	Promotion	HCC	[28]
ST6GAL1	TGF-β	Promotion	Breast cancer	[65]
ST6GAL1	EGFR	Promotion	Pancreatic cancer	[66]
ST6GAL1	PI3K/AKT/GSK3β/βcat	Promotion	Prostate cancer	[42]
ST6GAL1	Notch/Hes1/MMPs	Promotion	NSCLC	[68]
ST6GAL1	FGFR1-ERK/FAK	Promotion	Ovarian cancer	[69]
ST6GAL1	KAI1	Inhibition	Colon cancer	[71]
ST6GAL1	ICAM1	Inhibition	Colon cancer	[72]
ST6GAL2	ICAM1, VCAM1, MMP	Promotion	Breast cancer	[43]
ST6GALNAC1	sTn-CD44/MUC1	Promotion	Gastric cancer	[73]
ST6GALNAC1	IGF-1/STAT5b	Promotion	Gastric cancer	[74]
ST6GALNAC1	PI3K/AKT/NF-kB	Promotion	HCC	[44]
ST6GALNAC2	Galectin-3	Inhibition	Breast cancer	[76]
ST6GALNAC2	PI3K/AKT/NF-kB	Promotion	Breast cancer	[77]
ST6GALNAC2	PI3K/AKT/NF-kB	Promotion	Thyroid carcinoma	[78]
ST8SIA1	cMet	Promotion	Breast cancer	[79]
ST8SIA1	FAK-AKT-mTOR	Promotion	TNBC	[80]
ST8SIA4		Promotion	Breast cancer	[34]
ST8SIA4		Inhibition	Thyroid carcinoma	[35]
ST8SIA6-AS1	p38	Promotion	Breast cancer	[47]
ST8SIA6-AS1	DCAF4L2	Promotion	HCC	[46]
ST8SIA6-AS1	miR-125a-3p/NNMT	Promotion	Lung adenocarcinoma	[36]

**Table 3 cancers-13-02014-t003:** Sialyltransferases involved in immune evasion.

STs	Target/Mediator	Effect	Cancer Type	References
ST3GAL1	CD55	Promotion	Breast cancer	[92]
ST3GAL1/4	IL-10/6	Promotion	PDAC	[93]
ST3GAL3		Promotion	Melanoma	[90]
ST6GAL1	CD147/MMPs	Promotion	HCC	[91]

**Table 4 cancers-13-02014-t004:** Sialyltransferases involved in regulation of apoptosis and cell death.

STs	Target/Mediator	Effect	Cancer Type	References
ST6GAL1	FasR	Inhibition	Colon cancer	[95]
ST6GAL1	TNFR1	Inhibition	Pancreatic & ovarian cancer	[96]
ST6GAL1	TNF	Inhibition	Gastric cancer	[97]

**Table 5 cancers-13-02014-t005:** Sialyltransferases involved in angiogenesis.

STs	Target/Mediator	Effect	Cancer Type	References
ST3GAL1	Vasorin-TGFβ1	Promotion	Breast cancer	[100]
ST6GAL1	PECAM/VEGFR2/Intβ3	Promotion	Lewis lung carcinoma	[101]
ST6GAL1	VEGF	Promotion	Osteosarcoma	[102]
ST6GAL1	HIF-1α	Promotion	Ovarian, pancreatic cancers	[104]
ST8SIA1	EGF/VEGF	Inhibition	Pancreatic cancer	[107]

**Table 6 cancers-13-02014-t006:** Sialyltransferases involved in chemoresistance.

STs	Target/Mediator	Effect	Cancer Type	References
ST3GAL1		Promotion	ER+ breast cancer	[41]
ST3GAL1		Promotion	Ovarian cancer	[57]
ST3GAL1		Promotion	CML	[114]
ST3GAL4/6		Promotion	Gastric adenocarcinoma	[116]
ST6GAL1	EGFR	Promotion	Colon, ovarian cancers	[108,109]
ST6GAL1	FGFR1	Promotion	Ovarian cancer	[69]
ST6GAL1	HER2-AKT-ERK	Promotion	Gastric cancer	[111]
ST6GALNAC1	sTn	Promotion	Gastric cancer	[119]
ST6GALNAC2		Promotion	Colorectal cancer	[120]
ST8SIA4	P-gp/MRP1	Promotion	AML	[117]

**Table 7 cancers-13-02014-t007:** Sialyltransferases involved in stemness.

STs	Target/Mediator	Effect	Cancer Type	References
ST3GAL1	OLIG2, SALL2, SOX2	Promotion	Glioblastoma	[123]
ST6GAL1	SOX9, Slug	Promotion	Ovarian, pancreatic cancer	[124]
ST6GALNAC1	AKT/Galectin-3	Promotion	Colorectal cancer	[125]
ST8SIA1	FAK-AKT-mTOR	Promotion	TNBC	[80]

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
