# Peer review of "Aberrant Sialylation in Cancer: Biomarker and Potential Target for Therapeutic Intervention?"

_cancers, 2021, doi:10.3390/cancers13092014_

Round 1

Reviewer 1 Report

Manuscript ID: cancers-1156586 

Aberrant sialylation in cancer: biomarker and potential target 2 for therapeutic intervention?

Silvia Pietrobono and Barbara Stecca

It is a very well-written review. My comments are below

  • Aberrant Glycosylation involved both Fucosylation and Sialyation. Sialyation is reviewed in this article. In some cancers, fucosylation changes under precancerous conditions. Alpha L-fucosidase is a marker. Although the review's focus is on sialylation, it is good to mention the glycosylation process involves both fucosylation and sialyation. Evaluation is also done on fucosylation and sialyation in the serum and salivary samples of patients.
  • Elevated mRNA levels of ST3GAL2 and ST3GAL3 have a role in disease progression and metastasis of oral carcinoma. Its implications in Oral cancers are not mentioned. Please add  
  • A schematic diagram of sialylated glycans and their role in cancer progression may help since it is a major hallmark of cancer (discretionary)

Author Response

It is a very well-written review.

Response: We thank the Reviewer for the positive comments.

1. Aberrant Glycosylation involved both Fucosylation and Sialyation. Sialyation is reviewed in this article. In some cancers, fucosylation changes under precancerous conditions. Alpha L-fucosidase is a marker. Although the review's focus is on sialylation, it is good to mention the glycosylation process involves both fucosylation and sialyation. Evaluation is also done on fucosylation and sialyation in the serum and salivary samples of patients.

Response: We thank the Reviewer for the suggestion. We now mentioned that the most widely occurring cancer associated changes in glycosylation are sialylation, fucosylation, O-glycan truncation, and N- and O-linked glycan branching (lines 39-41).

2. Elevated mRNA levels of ST3GAL2 and ST3GAL3 have a role in disease progression and metastasis of oral carcinoma. Its implications in Oral cancers are not mentioned. Please add.

Response: We thank the Reviewer for the suggestion. In the revised version of the manuscript, we mentioned that elevated mRNA levels of ST3GAL2 and ST3GAL3 may have a role in oral squamous cell carcinoma progression and added the reference (lines 602-605).

3. A schematic diagram of sialylated glycans and their role in cancer progression may help since it is a major hallmark of cancer (discretionary).

Response: We have tried to draw a schematic diagram of sialylated glycans and their role in cancer progression, as suggested by the Reviewer. However, the diagram got too complex and busy to be useful and we decided to not include it.

Reviewer 2 Report

This review article explored the mechanisms by which sialyltransferases promote cancer progression, pointed out that the possible use of sialyltransferases as biomarkers for cancer, and summarize the most promising findings on the development of sialyltransferase inhibitors as potential anti-cancer treatments.

It is straightforward, well design, well written, and concise and has clear results. Definitely deserves to be published and is a valuable contribution to the Cancers. Some minor issues could be addressed before publication.

Minor points:

There are some opposite opinions for the same sialyltransferases in your article, such as:

  1. In the part ‘4.2. Activating invasion and metastasis’ (line 265-273), ST6GAL1 showed tumor suppressing role in colorectal cancer. However, in the part ‘4.4. Evading apoptosis and cell death’ (line 409-412), ST6GAL1-mediated sialylation of FasR blocked Fas internalization and the formation of the death-inducing signaling complex, disabling apoptotic signaling in colon cancer cells, which can promote tumor. What’s your opinion about this opposition?
  2. In the part ‘4.1. Sustaining proliferation and tumor growth’, a study revealed a positive feedback loop between ST3GAL1 and GDNF/GFRA1(glial cell derived neurotrophic factor) signaling in breast cancer cells (line 162-167). GDNF also expressed in gliomas, however, in gliomas increased expression of STs may have a tumor inhibitory effect (line 189-196). What’s your opinion about this?

Author Response

It is straightforward, well design, well written, and concise and has clear results. Definitely deserves to be published and is a valuable contribution to the Cancers. Some minor issues could be addressed before publication.

Response: We thank the Reviewer for the positive comments.

There are some opposite opinions for the same sialyltransferases in your article, such as:

1. In the part ‘4.2. Activating invasion and metastasis’ (line 265-273), ST6GAL1 showed tumor suppressing role in colorectal cancer. However, in the part ‘4.4. Evading apoptosis and cell death’ (line 409-412), ST6GAL1-mediated sialylation of FasR blocked Fas internalization and the formation of the death-inducing signaling complex, disabling apoptotic signaling in colon cancer cells, which can promote tumor. What’s your opinion about this opposition?

Response: The opposite functions of ST6GAL1 in colon cancer (tumor suppressive role in invasion, pro-tumorigenic role in Fas-mediated apoptosis) could be due to a dynamic change in the expression of ST6GAL1 and its substrates during CRC progression, suggesting distinct roles of ST6GAL1 at different stages of the disease (Zhang S, BBRC, 2017).

2. In the part ‘4.1. Sustaining proliferation and tumor growth’, a study revealed a positive feedback loop between ST3GAL1 and GDNF/GFRA1(glial cell derived neurotrophic factor) signaling in breast cancer cells (line 162-167). GDNF also expressed in gliomas, however, in gliomas increased expression of STs may have a tumor inhibitory effect (line 189-196). What’s your opinion about this?

Response: We thank the reviewer for this comment. Although increase in sialyltransferase expression promotes pro-metastatic and pro-tumorigenic effects in the majority of cancer types, it can have opposite effect on cancers originating from neural tissues, such as glioma, possibly due to the fact that polysialylation plays a key role in adult brain plasticity and regeneration. This comment has been added in the text (lines 199-203).

Reviewer 3 Report

The article entitled “Aberrant sialylation in cancer: biomarker and potential target for therapeutic intervention?” by Pietrobono and Stecca reviews the role of aberrant sialylation in cancer progression through enhancing migration, invasion and angiogenesis in several types of cancers. It is specially focused on the importance of sialyltransferases in promoting cancer progression. The authors deftly summarize the most recent advances on the regulation of sialylation expression in cancer and detail the sialyltransferases and the mediators that are involved in the eight events of tumour progression (Sustaining proliferation, activating invasion and metastasis, promoting immune evasion, evading apoptosis, inducing angiogenesis, promoting chemoresistance, enhancing stemness and inducing epithelial mesenchymal transition).

The topic is really important and interesting because it also address the compounds that can function as sialyltransferases inhibitors and that could be potentially used in antitumoral treatments.

I enjoyed reading the manuscript and I think the review work was well-performed, the manuscript is well-written and includes the most important findings on this topic, but currently the paper has small matters that should be addressed,

Concerns:

  1. I am aware that N-acetilneuraminic acid is the most common sialic acid found in outer terminal glycans and that N-glycolylneuraminic acid (Neu5Gc) is not expressed in humans in normal situation due to the deletion in the CMAH gene that codify the hydroxylase that converts CMP-Neu5Ac to CMP-Neu5Gc. However, it would be interesting to mention in the manuscript, that aberrant glycosylation in cancer can also include Neu5Gc sialoglycans and that both, this sialic acid and anti-Neu5Gc antibodies have been also proposed as cancer biomarkers.
  2. It is known that there is a clear link between chemoresistance of tumours and hypoxia, thus it would be interesting to include the bibliography related with the effect of hypoxia in sialylation profiles of ovarian and breast cancer cell lines when discussing the role of ST3GAL4.
  3. To improve the manuscript, I would suggest organizing information about the roles of sialyltransferases in cancer (eight items included in Figure 1) according to the order they are presented in the text (section 4, line 153). The hallmark inducing EMT appears in the Figure but it has not been considered as a sub-section in paragraph 4. The inducing EMT information is discussed in section 4.2 (line 203) and, therefore, the tittle of this subsection could include both, activating invasion and metastasis and EMT inducing events for better understanding.
  4. Heading of section 5 refers to sialyltransferases as cancer biomarkers, however all the glycoproteins described in the first paragraph are glycoproteins used as a cancer biomarkers in different types of tumours. I would suggest including sialylated glycoproteins in the headline “Sialylated glycoproteins and sialyltransferases as cancer biomarkers”.
  5. Other English spelling mistakes should be addressed:

- sialic acid instead of acid sialic (lines 14, 46, 59, 349, 358)

- siglec-10 instead of siglec10 (lines 370 and 371)

- glycosylated proteins instead glycosylation proteins (line 588)

Author Response

The topic is really important and interesting because it also address the compounds that can function as sialyltransferases inhibitors and that could be potentially used in antitumoral treatments.

I enjoyed reading the manuscript and I think the review work was well-performed, the manuscript is well-written and includes the most important findings on this topic, but currently the paper has small matters that should be addressed

Response: We thank the Reviewer for the very positive comments on our manuscript.

1. I am aware that N-acetilneuraminic acid is the most common sialic acid found in outer terminal glycans and that N-glycolylneuraminic acid (Neu5Gc) is not expressed in humans in normal situation due to the deletion in the CMAH gene that codify the hydroxylase that converts CMP-Neu5Ac to CMP-Neu5Gc. However, it would be interesting to mention in the manuscript, that aberrant glycosylation in cancer can also include Neu5Gc sialoglycans and that both, this sialic acid and anti-Neu5Gc antibodies have been also proposed as cancer biomarkers.

Response: We thank the reviewer for this suggestion. In the revised version of the manuscript we discussed that anti-Neu5Gc antibodies have been proposed as possible cancer biomarkers (lines 615-625).

2. It is known that there is a clear link between chemoresistance of tumours and hypoxia, thus it would be interesting to include the bibliography related with the effect of hypoxia in sialylation profiles of ovarian and breast cancer cell lines when discussing the role of ST3GAL4.

Response: We thank the Reviewer for the suggestion. In the revised version of the manuscript we added the reference related to the effect of hypoxia in sialylation profiles of ovarian and breast cancer cell lines when discussing the role of ST3GAL4 (lines 466-469).

3. To improve the manuscript, I would suggest organizing information about the roles of sialyltransferases in cancer (eight items included in Figure 1) according to the order they are presented in the text (section 4, line 153). The hallmark inducing EMT appears in the Figure but it has not been considered as a sub-section in paragraph 4. The inducing EMT information is discussed in section 4.2 (line 203) and, therefore, the title of this subsection could include both, activating invasion and metastasis and EMT inducing events for better understanding.

Response: We changed the title of subsession 4.2 to “Activating invasion and metastasis, and EMT inducing events” (lines 212) for better clarity, as suggested.

4. Heading of section 5 refers to sialyltransferases as cancer biomarkers, however all the glycoproteins described in the first paragraph are glycoproteins used as a cancer biomarkers in different types of tumours. I would suggest including sialylated glycoproteins in the headline “Sialylated glycoproteins and sialyltransferases as cancer biomarkers”.

Response: We thank the Reviewer for this suggestion. We changed headline to “Sialylated glycoproteins and sialyltransferases as cancer biomarkers” (line 567).

5. Other English spelling mistakes should be addressed:

- sialic acid instead of acid sialic (lines 14, 46, 59, 349, 358).

Response: This has been fixed throughout the manuscript.

- siglec-10 instead of siglec10 (lines 370 and 371):

Response: This has been fixed throughout the manuscript.

- glycosylated proteins instead glycosylation proteins (line 588).

Response: We apologize for the mistake, we now corrected it (line 607).